# Alpha Radiation as a Way to Target Heterochromatic and Gamma Radiation-Exposed Breast Cancer Cells

**DOI:** 10.3390/cells9051165

**Published:** 2020-05-08

**Authors:** Maja Svetličič, Anton Bomhard, Christoph Sterr, Fabian Brückner, Magdalena Płódowska, Halina Lisowska, Lovisa Lundholm

**Affiliations:** 1Department of Molecular Biosciences, The Wenner-Gren Institute, Stockholm University, 106 91 Stockholm, Sweden; maja.svetlicic@gmail.com (M.S.); toni.bomhard@googlemail.com (A.B.); csterr@bfs.de (C.S.); fabbr08@t-online.de (F.B.); 2Department of Radiobiology and Immunology, Institute of Biology, Jan Kochanowski University, 25-406 Kielce, Poland; magdalena.anna.plodowska@gmail.com (M.P.); halina.lisowska@ujk.edu.pl (H.L.)

**Keywords:** alpha radiation, gamma radiation, chromatin, DNA damage, breast cancer

## Abstract

Compact chromatin is linked to a poor tumour prognosis and resistance to radiotherapy from photons. We investigated DNA damage induction and repair in the context of chromatin structure for densely ionising alpha radiation as well as its therapeutic potential. Chromatin opening by histone deacetylase inhibitor trichostatin A (TSA) pretreatment reduced clonogenic survival and increased γH2AX foci in MDA-MB-231 cells, indicative of increased damage induction by free radicals using gamma radiation. In contrast, TSA pretreatment tended to improve survival after alpha radiation while γH2AX foci were similar or lower; therefore, an increased DNA repair is suggested due to increased access of repair proteins. MDA-MB-231 cells exposed to fractionated gamma radiation (2 Gy × 6) expressed high levels of stem cell markers, elevated heterochromatin H3K9me3 marker, and a trend towards reduced clonogenic survival in response to alpha radiation. There was a higher level of H3K9me3 at baseline, and the ratio of DNA damage induced by alpha vs. gamma radiation was higher in the aggressive MDA-MB-231 cells compared to hormone receptor-positive MCF7 cells. We demonstrate that heterochromatin structure and stemness properties are induced by fractionated radiation exposure. Gamma radiation-exposed cells may be targeted using alpha radiation, and we provide a mechanistic basis for the involvement of chromatin in these effects.

## 1. Introduction

Resistance of tumours to radiotherapy and chemotherapy is a severe problem. Breast cancer is among the most common cancers in women, and although survival has improved the last decades, the heterogeneity of the disease is large [1]. The prognosis is more favourable for the luminal subtypes, while the human epidermal growth factor receptor 2 (HER2) and basal-like subtypes recur more frequently and the overall survival is lower [1]. Half of all breast cancer patients receive photon radiotherapy, commonly as adjuvant therapy after surgery, although the HER2 and triple negative subtypes are reported to respond less well than luminal cancers [1]. Apart from intrinsic factors such as DNA repair capacity that may influence cellular radiosensitivity [2], it is now realised that, in cases where cell killing is insufficient, tumour cells acquire a radioresistant phenotype as a consequence of fractionated irradiation [3,4]. The mechanisms of this acquired radioresistance are not fully understood.

The breast cancer cell lines MDA-MB-231 and MCF7 are well characterised and used in numerous in vitro studies aiming to improve breast cancer therapy response. MDA-MB-231 is a triple negative cell line; negative for estrogen receptor α (ER), progesterone receptor (PR), and HER2; and classified as a basal subtype, while the ER+/PR+ MCF7 is designated as a luminal A subtype [5]. Previous studies have also reported that the MDA-MB-231 cells are more radioresistant than MCF7 [6].

Chromatin structure plays an important role in the processes of DNA metabolism. It is dynamic and generally classified into open, transcriptionally active euchromatin and more condensed, transcriptionally inactive heterochromatin, ranging from facultative to constitutive heterochromatin [7]. Chromatin structure is regulated by a balance between the condensing histone deacetylases (HDACs) and the opening histone acetyltransferases (HATs). Many HDAC inhibitors (HDACi) have been developed, and when the deacetylases are blocked, the histone tails become hyperacetylated, a sign of an open, euchromatic structure. It is assumed that the tight packaging of DNA is responsible for a radioresistant phenotype, but the full relationship between chromatin structure and response of cells to cancer treatment is not understood.

Ionising radiation is divided into low linear energy transfer (LET) radiation such as gamma radiation/X-rays and high LET radiation (>10 keV/µm) such as alpha particles/heavy ions. Low LET radiation mainly damages DNA indirectly via free radicals formed from water radiolysis, while direct ionisation of atoms in DNA dominate after high LET radiation [8]. High LET radiation causes clustered DNA damage along its track, resulting in a high level of complex damage with delayed repair, represented by a slower removal of γH2AX [9,10] and 53BP1 foci [11] as compared to low LET radiation. Alpha radiation generally has a higher relative biological effectiveness (RBE) than gamma radiation due to the induction of clustered damage [12]. The RBE value depends on the assay and cell system but is commonly between 1 and 13; for clonogenic cell survival, it is typically around 4 [13].

Condensed chromatin has been shown to be less susceptible to damage induction by gamma radiation [14]. When comparing gene-poor regions with low transcriptional activity to highly transcribed regions, not only the chromatin compactness but also the higher abundance of proteins were suggested to play a role [14]. Using isolated nuclei, the frequency of double strand breaks (DSB) was lower in condensed versus decondensed chromatin [15]. Heterochromatic structure also restricts DNA damage response signalling and repair, and induced DSBs are relocated to the heterochromatin–euchromatin border [16]. The role of chromatin has not been thoroughly studied in response to high LET damage.

Besides the higher biological efficiency, alpha radiation has several other advantages compared to conventional gamma radiation therapy. It was shown already in 1966 that, the higher the LET, the lower the dependence on the oxygen level inside the tumour [17]. The oxygen enhancement ratio, relating the dose needed for the same biological response in hypoxic cells versus normoxic cells, is lower for high LET radiation [18]. Additionally, alpha emitters are reported to act independently of proliferation, making them ideal for targeting stem-like cells [19].

The high induction of DNA damage close to the particle emitter is advantageous clinically, where surrounding normal tissue is spared due to the short range of alpha particles (50–100 µM) [19]. An alpha emitter which selectively binds to bone metastases of castration-resistant prostate cancer (Xofigo, ^223^RaCl_2_) was approved in 2013, producing improved overall survival and fewer adverse events than placebo [20]. Other alpha emitters such as ^213^Bi, ^221^At, and ^225^Ac have been included in clinical trials for glioma, ovarian carcinoma, metastatic melanoma, and acute myeloid leukaemia, in most cases bound to monoclonal antibodies towards tumour-specific proteins (reviewed in References [19,21,22,23]). So far, the other approved radionuclide therapies are β-emitters which can reach up to 2 mm in tissue. Yet, due to the comparatively insufficient amounts of radiation delivered and the difference in LET, a higher efficacy is indicated for alpha emitters compared to beta emitters [24].

In this study, we first wanted to evaluate the relative importance of DNA damage induction and repair after low LET gamma irradiation and high LET alpha irradiation. By analysing cells with an open chromatin at the time of exposure (HDACi pretreatment), we aimed to mechanistically explore the impact of chromatin structure on the response of cells to the different radiations using clonogenic survival and γH2AX foci analysis. Secondly, the response to alpha radiation was analysed in breast cancer models. By applying two different fractionation schemes, we established gamma radiation-exposed cells which were stem-like, heterochromatic, and had an improved response towards alpha radiation. The same properties were not seen in the hormone-sensitive MCF7 cells after culture in sphere-forming media. A higher ratio of DNA damage induction by alpha vs. gamma radiation was shown in the aggressive MDA-MB-231 cells, which may be explained by a higher basal level of a heterochromatin marker and suggests a possible use of alpha emitters to target heterochromatic, radiotherapy-exposed cells.

## 2. Materials and Methods

### 2.1. Cell Culture

The human breast cancer cell line MDA-MB-231 was cultured in Dulbecco’s Modified Eagle Medium (DMEM) containing 10% defined bovine serum, and MCF7 was grown in Eagle’s minimum essential medium supplemented with 2 mM L-glutamine and 10% defined bovine serum, with 1% penicillin-streptomycin added for irradiation experiments (Sigma-Aldrich, Stockholm, Sweden). The cell lines were from Sigma-Aldrich and were maximally grown for 2.5 months/20 passages, and their mycoplasma-negativity was confirmed by the lack of cytoplasmic dots using immunofluorescence with 4′,6-diamidino-2-phenylindole (DAPI) staining.

To perform the first setup of fractionated exposure in MDA-MB-231, 6 × 0.5/1.0/1.5 Gy (3 days per week), cells were seeded at 1,000,000 (A) or 1,500,000 (B) cells/25 cm^2^-flask in the mornings of days 0, 2, 4, 7, 9, and 11 and irradiated in the afternoons after cell attachment (>3 h after); 1,000,000/1,500,000 cells were always reseeded for A and B respectively, and the remaining cells were pelleted for RNA analysis. At day 14, cells were collected for RNA analysis and seeded for micronuclei preparation.

To produce the radiation-exposed (RE) subline of MDA-MB-231, 500,000 (control 1, RE 1) or 1,000,000 cells (control 2, RE 2) were seeded per 25 cm^2^-flask. This time, 6 × 2 Gy (2 days per week) was given on days 1, 3, 8, 10, 15, and 17. After the first week, the cell number was adjusted to keep an appropriate number of cells, with higher numbers put back for RE cells. At 18 days after the last dose, cells were frozen in −150 °C. Control 1 and RE 1 cells were used for experiments at 4–6 weeks after 12 Gy was reached (freezing periods excluded). Cells were seeded for clonogenic survival assay as in Section 2.4 and re-irradiated with single doses of gamma or alpha radiation, and cell pellets were collected for RNA and protein analysis.

To form MCF7 tumour-initiating cells (TICs), cells were cultured in suspension (seeded at 10,000 cells/mL) for 10 days in serum-free DMEM/F12 medium supplemented with epidermal growth factor (EGF), basic fibroblast growth factor (bFGF), heparin, and 1xB27 supplement, using an established protocol [25,26,27].

### 2.2. Western Blot

After cell harvesting by trypsination and freezing of cell pellets, proteins were extracted in radioimmunoprecipitation assay (RIPA) buffer and SDS-PAGE/Western blot was performed as previously described [27]. Primary antibodies used were acetylated lysine 8 of histone H4 (H4K8ac, Cell Signaling Technology, Danvers, MA, USA), trimethylated lysine 9 of histone H3 (H3K9me3, Abcam, Cambridge, UK), histone H3 (Millipore/Merck, Darmstadt, Germany), and glyceraldehyde 3-phosphate dehydrogenase (GAPDH) (Sigma-Aldrich).

### 2.3. Irradiation

Alpha exposure was performed using an ^241^Am source with a dose rate of 0.223 Gy/min and an average LET of 91 keV/µm, as detailed in Reference [28]. Cells were plated on glass cover slips placed in 6-well plates. Cover slips were transferred to a polyamide disc, covered with a 2.5-µm thick Mylar foil to prevent evaporation, and positioned in close contact with the source using a motor device. Gamma irradiation was performed using a Gammacell 40 Exactor (MDS Nordion, Ontario, Canada) ^137^Cs source with a dose rate of 0.79 Gy/min.

### 2.4. Clonogenic Survival Assay

MDA-MB-231 and MCF7 cells were plated at a density of 200 cells/well for gamma radiation and 400 cells/well for alpha radiation (due to cell loss when removing the Mylar foil). MCF7 TICs were seeded at 400 or 800 cells/well, respectively. For trichostatin A (TSA) pretreatments, cells were plated in the morning and treated with TSA in the afternoon. The next day, cell medium was replaced at the time of radiation exposure and colonies were allowed to form for 14 days (except control and RE, which were stained at 8–10 days following exposure). Staining was performed with 5% Giemsa in 25% methanol. Colonies with at least 50 cells were manually counted for MDA-MB-231, while semiautomatic counting was performed for MCF7 using the ImageJ macro countPHICS [29].

### 2.5. γH2AX Immunofluorescence and Scoring

Cells were seeded on cover slips at least 2 days prior to irradiation and were fixed in 70% EtOH for 10 min at the indicated time points after exposure. Cells were permeabilised in 0.2% Triton-X in phosphate-buffered saline (PBS) and incubated with anti-phospho-Histone H2AX antibody Ser139 (Upstate/Millipore) followed by secondary anti-mouse IgG fluorescein isothiocyanate (FITC) (Sigma-Aldrich), both in PBS containing 2% bovine serum albumin (BSA). Coverslips were counterstained using DAPI and mounted on an objective glass in Vectashield. Photos were taken, and the total number of γH2AX foci per cell was scored in 50 cells per treatment and experiment using a macro for ImageJ version 1.43u, as previously described [30].

### 2.6. Real-Time PCR

RNA preparation was performed using the E.Z.N.A. Total RNA Kit I (Omega Bio-tek, Norcross, GA, USA). cDNA was synthesised using random hexamer primers with the High-Capacity cDNA Reverse Transcription Kit (Thermo Fisher Scientific, Göteborg, Sweden). Primers were towards Sox2, Oct4, Nanog, CD133, 18S (sequences in References [22,23] and CD44; forward: 5′-GTCGAAGAAGGTGTGGGCAGAA, reverse: 5′-AAATGCACCATTTCCTGAGACTTGCT) (LGC Biosearch Technologies, Risskov, Denmark). Reaction mixes containing primers, cDNA, and 5x HOT FIREPol^®^ EvaGreen^®^ qPCR Supermix (Solis BioDyne, Tartu, Estonia) were run in duplicate on a LightCycler^®^ 480, starting at 95 °C for 15 min, followed by 40 cycles of 95 °C for 15 s, 60 °C for 20 s, and 72 °C for 20 s. The 2^−ΔΔCt^ method was used, and primer specificity was confirmed using melting curve analysis.

### 2.7. Micronucleus Assay

Cells were reseeded, and cytochalasin B (final concentration 5 mg/mL; Sigma-Aldrich) was added on day 14 (3 days after the last fraction) and kept for 24 h. Cells were detached using trypsin; otherwise, harvesting was described previously [26]. The fixed cells were dropped on clean slides and stained with Giemsa the next day. The presence of micronuclei (MN) was scored in a blinded manner in 1000 binucleated cells (BNCs) per sample, using a light microscope with 400× magnification.

### 2.8. Statistics and Curve Fits

Results are displayed as mean ± standard deviation from at least 3 experiments, except for Figures 3 and 4B,C where 1–2 replicates are shown. Statistical differences between two groups were analysed using an unpaired, two-tailed Student’s *t*-test. For Figure 1C, an ANOVA plus the Bonferroni’s multiple comparison test was applied. Analysis of survival curves was performed in GraphPad Prism v5.00 (GraphPad Software, San Diego California USA) and fitted by entering the linear-quadratic equation [31] Y = exp(−1*(A*X + B*X^2)) for gamma radiation, using the least-square method with the initial value of A set to −1.0 and B set to −0.1, according to instructions in frequently asked questions (FAQ). A and B represent fitting coefficients, Y is the survival fraction, and X is the dose in Gy. For alpha radiation, a linear function [32] was fitted using the exponential one phase decay, where the plateau was constrained to 0 and Y0 was constrained to 1. Significant differences were analysed based on the fitted curves in Figure 1C. Straight lines were also fitted to the dose response data in Figure 2A. In all graphs, symbols are nudged for transparency.

## 3. Results

### 3.1. Effects of Trichostatin A (TSA) on Chromatin Structure and Clonogenic Survival after Exposure to Gamma vs. Alpha Radiation

To evaluate the role of chromatin in response to high or low LET radiation, we first aimed to certify that we have an open chromatin at the time of exposure. Using TSA doses ranging from 0.25–1 µM, we assessed acetylated lysine 8 of histone H4 (H4K8ac) as an indirect measure of euchromatin in MDA-MB-231. At 18 h after exposure, a 0.5 µM dose of TSA produced a detectable band, while the highest dose of 1 µM TSA gave the most pronounced increase in H4K8ac (Figure 1A); 0.5 or 1 µM of TSA alone did not induce prominent decreases in clonogenic survival (Figure 1B). To investigate the net effect on survival using TSA pretreatment, we analysed the response of MDA-MB-231 cells to alpha and gamma radiation. The highest radiation doses were selected to induce a similar level of survival (ca 20–30%). Pretreatment with 1 µM, but not 0.5 µM of TSA, sensitised the cells to gamma radiation (Figure 1C). In contrast, both doses of TSA pretreatment had the opposite effect in response to alpha radiation, where survival was improved.

### 3.2. Formation and Removal of γH2AX Foci in TSA-Pretreated MDA-MB-231 Exposed to Gamma and Alpha Radiation

The survival data suggest that the most important effects of TSA pretreatment is an enhancement of DNA damage induction in response to gamma radiation, while an improved DNA repair could be central for the reaction to alpha particles. To further dissect this, we evaluated the effects of TSA pretreatment on the formation of foci of the DSB marker γH2AX. We first analysed the dose response after alpha and gamma radiation at 30 min after exposure where the DNA damage induction is highest. For gamma, we noted an increased response at all tested doses; for alpha, the increase was most prominent for the higher doses: 0.75 and 1 Gy (Figure 2A). One reason for the higher dose needed to induce γH2AX foci above the control level and the generally lower level of γH2AX foci for alpha radiation than gamma radiation is the difference in distribution of hits per cell nucleus. Per unit dose, fewer cells are hit by alpha particles than by gamma radiation [12]. In addition, alpha tracks are traversing cells from several angles, and we know that we generally underestimate the number of foci in cases where the track goes vertically through the cells due to our setup for immunofluorescence microscopy (single plane, not confocal microscopy).

Doses of 2 and 0.75 Gy for gamma and alpha, respectively, were selected for repair kinetics analysis. Those doses induced detectable increases in foci number compared to control while still staying at a submaximal level to allow for potential increases using TSA. We detected a higher number of γH2AX foci upon chromatin opening by TSA during the first 30 min after gamma radiation, with significantly elevated levels only at 15 min after exposure (Figure 2B). The pattern was clearly different after alpha radiation. The number of γH2AX foci was highest in the TSA-treated samples at 15 min after exposure, which might be attributed mainly to a faster response in euchromatic TSA-treated cells, while the damage induction peaked at 30 min in the relatively heterochromatic MDA-MB-231 without TSA pretreatment. Interestingly, from 30 min and up to 4 h, the γH2AX focus number tended to be lower after TSA pretreatment, and values were significantly lower at 2 h. Generally, most gamma radiation-induced damage appeared repaired within 1 h, indicative of damage, which is easier to repair compared to clustered, alpha radiation-induced damage. Alpha radiation produced a biphasic response, which is reported to be a sign of the fast and slow DNA repair processes, where heterochromatic DSBs are repaired by resection-dependent slow processes in G1 and G2 [16,33]. Therefore, it was notable that the second γH2AX peak after alpha exposure was removed by TSA-pretreatment, suggesting an improved repair, even better than at baseline for TSA-pretreated samples.

There was an elevated level of γH2AX foci by pretreatment with 1 µM TSA only (0 h), most likely reflecting some degree of toxicity as displayed in Figure 1B. Consequently, control values were subtracted to allow for comparisons with or without TSA in Figure 2. The same graphs without subtraction of controls are displayed in Appendix A.

The foci areas were also analysed after classification into small and large foci [11]; their distinction using a threshold of 60 pixels is shown in Figure 2C. It is more difficult for the cell to repair complex, clustered damage, defined by >2 lesions within 1–2 turns of the DNA helix [34], such as in large foci. Alpha or other high LET radiation induce more large foci [9,35], as seen in Figure 2C. The small foci (Figure 2D) display a similar pattern as for the total foci number. Large foci, in particular, tended to be repaired better after TSA pretreatment at 3–4 h after alpha radiation exposure, in line with data for all foci, while the initial early peak at 15 min also contained a substantial proportion of large foci (Figure 2E). Due to the noticeably different response in clonogenic survival and γH2AX foci number between gamma and alpha radiation, we suggest that DSB induction is prominent for gamma but is of minor importance for the action of alpha radiation in TSA-pretreated cells. Open chromatin allows for better access of repair proteins and an improved DNA damage repair, and this appears to be the major factor influencing the increased survival after alpha radiation of cells with open chromatin.

### 3.3. Fractionated Gamma Radiation Exposure in MDA-MB-231 Cells

Two different fractionation schemes were tested for MDA-MB-231 cells. The first setup of fractionated exposure consisted of six fractions of 0.5, 1.0, or 1.5 Gy during 3 days per week to establish the level of fractionated radiation that the cells could handle for longer term culture and to characterise their response; 1 million or 1.5 million cells were always reseeded for A and B, respectively, and both cell numbers produced dose-dependent reductions in cell growth (Figure 3A). There was also a dose-dependent increase in the formation of micronuclei (Figure 3B), which are chromosomes or acentric fragments lagging behind at anaphase and not incorporated into the nucleus [36]. RNA levels of cancer stem cell markers CD44 and CD133 as well as normal stem cell markers Sox2, Oct4, and Nanog started to increase towards day 14, mainly in the 1.5 Gy/fraction samples (Figure 3C). Based on experiences from this first fractionation experiment, a second fractionation setup was applied where we used higher doses and longer gaps aiming to increase stem cell marker expression; therefore, six fractions of 2 Gy were given twice per week.

### 3.4. Cellular Properties and Response to Alpha Radiation in Gamma Radiation-Exposed MDA-MB-231

To further evaluate the role of alpha or high LET radiation in the preclinical context, we aimed to evaluate the efficacy of alpha radiation in cells exposed to fractionated gamma radiation, reminiscent of the radiotherapy setup. Using the current setup (6 × 2 Gy) with a higher dose followed by longer gaps (Figure 4A), we achieved a prominent reduction of cell growth and an increase in normal and cancer stem cell markers starting already at a total dose of 6 Gy (Figure 4B,C).

Surviving cells after 12 Gy were allowed to regrow and were subsequently used after 4–6 weeks of culture as illustrated in Figure 4A. The cancer stem cell markers CD44 and CD133, and the normal stem cell markers Sox2, Oct 4, and Nanog were all increased at this time point (Figure 4D). In concordance with a more tightly packed chromatin structure of stem-like cells, the heterochromatin marker trimethylated lysine 9 of histone H3 (H3K9me3) was increased as well (Figure 4E). There was however no resistance to gamma radiation at this late time point (Figure 4F).

Interestingly, the radiation-exposed cells were now more sensitive to alpha radiation at the majority of doses (Figure 4F). The area under the curve (AUC) was reduced by 33% (using the data points; GraphPad Prism) or 47% (using the fitted curves) in radiation-exposed cells versus control. This was assayed using clonogenic survival, the end point that sums up all cell death events. Taken together, these data indicate that heterochromatic stem-like gamma radiation-exposed MDA-MB-231 cells tend to be more sensitive to alpha radiation. Based on the results using the HDAC inhibitor TSA, we suggest that this phenomenon is mainly due to the ability of alpha radiation to go through chromatin and propose this as a biological basis for future evaluation of targeted alpha emitter therapy.

### 3.5. Analysis of Heterochromatin Structure and Radiation Response in Luminal Type MCF7 Compared to Triple Negative MDA-MB-231 Cells

Since the hormone-receptor positive cell line MCF7 belongs to the more favourable (in terms of therapy choices and survival) luminal A breast cancer subtype, and since heterochromatin structure has been linked to radioresistance and poor survival, we aimed to analyse if those cells differed in response compared to MDA-MB-231. MCF7 was grown as tumour-initiating cells (TICs) according to a protocol previously established for non-small cell lung cancer (NSCLC) TICs [25,26,27]. Cells were growing as clusters in suspension and had increased levels of CD44 but no increase in the other tested markers (Figure 5A). MCF7 TICs displayed no significant differences in H3K9me3 levels or in response to gamma or alpha radiation compared to MCF7 (Figure 5B,C).

When comparing the two breast cancer cell lines with different characteristics, the basal levels of H3K9me3 are much higher in MDA-MB-231 cells (Figure 6A). Although total histone H3 protein was expressed at a higher level in MDA-MB-231 compared to MCF7 as well, the difference was even larger for H3K9me3. These data fit with the reports of a link between heterochromatin markers and poor tumour prognosis. Additionally, we assessed their response to high doses of gamma and alpha radiation using γH2AX analysis at an early (30 min) and late (24 h) time point. The number of foci induced after 6 Gy of gamma radiation was higher in MCF7 cells than in MDA-MB-231 cells, while the foci number reached after 2 Gy alpha radiation was relatively similar in both cell lines (Figure 6B). The same pattern was seen when comparing the AUC; the difference in response to gamma (317 vs. 189; 1.7 fold) compared to alpha radiation (150 vs. 132: 1.1-fold) for MCF7 versus MDA-MB-231 cells is evident, translating into a 1.5-fold higher sensitivity relatively for alpha vs. gamma radiation in MDA-MB-231 versus MCF7 cells when comparing the alpha/gamma AUC ratios between the different cell lines (Appendix A). MDA-MB-231 had higher basal levels of γH2AX foci without ionising radiation (IR), which could be a sign of higher genomic instability. Also, when differences in basal levels were taken into account by subtracting control γH2AX values, there was a 1.3-fold higher sensitivity relatively for alpha vs. gamma in MDA-MB-231 vs. MCF7 (Appendix A). The cell lines could still perform repair similarly as judged by these time points; damage was almost down to base line at 24 h after gamma radiation, while γH2AX levels were staying nearly as high, illustrating the complexity of damage and lack of repair after this high alpha radiation dose.

## 4. Discussion

Various heterochromatin proteins (HP1α, β, or γ) were reported as markers of poor prognosis in sarcoma, breast, non-small cell, and prostate cancer [37,38,39,40]. We have previously shown that therapy-resistant NSCLC tumour-initiating cells displayed elevated levels of H3K9me3 and stem cell markers and could be sensitised to gamma radiation or cisplatin by HDACi pretreatment [25]. Because of the connection between a heterochromatic phenotype and poor tumour prognosis, we now analysed the epigenetic modification of histone H3 in two subtypes of breast cancer cells. Indeed, the more aggressive MDA-MB-231 cell line displayed higher levels of the H3K9me3 histone mark as compared to MCF7 cells. This is in line with data showing a correlation between epigenetic events and chromatin structure with cancer progression [41]. In future studies, it would also be of value to assess changes in chromatin structure using methods such as micrococcal nuclease assay or DNA staining.

Although chromatin structure has been shown to influence radiation response, most studies focused on gamma radiation, which markedly differs from alpha radiation in the mechanism of action. To the best of our knowledge, this is the first report analysing HDACi in connection with alpha radiation. We see a consistency between results obtained with clonogenic survival and γH2AX focus analysis, where the HDACi TSA markedly sensitised cells to gamma radiation but not to alpha radiation. The results using alpha radiation are more variable due to reasons such as a more challenging exposure procedure and the inherent properties of alpha radiation where not all cells are hit at lower doses. Future studies using other methods are needed; still, the pattern after alpha radiation is clearly different from that after gamma radiation. It would also be interesting to analyse downstream markers of non-homologous end joining (NHEJ) such as 53BP1 foci and of homologous recombination (HR; RAD51 foci) to further pinpoint the roles of the different repair pathways for high LET-induced complex/clustered DNA damage.

There are a few reports using HDACi together with high LET carbon ions with partly different results. In support of our data, the HDACi vorinostat/suberoylanilide hydroxamic acid (SAHA) had a radioprotective effect when combined with carbon ions in an osteoblast cell line, while the effect was opposite in sarcoma cell lines [42]. When vorinostat and carbon ions were combined in two glioblastoma cell lines, there was a slight sensitisation in cell viability, with a prolonged γH2AX foci only for one of the cell lines [43]. The HDACi cyclic hydroxamic-acid-containing peptide 31 (CHAP31) sensitised esophageal squamous cell carcinoma to carbon ions both in cell experiments and mouse xenografts, but it was also accompanied by a reduction in gene expression of the Mre11-Rad50-Nbs1 (MRN) complex, which is crucial for DSB repair [44]. The divergent results could depend on the nonhistone effects of the HDACi, the scheduling scheme, as well as cell line differences in expression of HDACs and other target proteins.

The DNA repair process involves reorganisation and decondensation of chromatin, but changes seem to be local and are not modifying heterochromatin-specific histone marks (H3K9me3, H4K20me3) at the pericentromere [45]. Using Monte Carlo simulations, cell nucleus models were generated representing heterochromatin, euchromatin, or a mix of both [46]. More direct and less indirect damages were observed in heterochromatin compared to euchromatin in response to protons or alpha radiation, which is consistent with the notion of heterochromatin as protective against the low LET-produced reactive species. DSB complexity was simulated to increase with LET, but interestingly, there was no difference in induced damage complexity between heterochromatin and euchromatin [46]. Instead, an inefficient DSB repair in heterochromatin indicated to play a role and foci cluster size, detected by transmission electron microscopy, was shown to increase by time after high LET radiation [47]. Therefore, an already open chromatin by TSA may very well modulate the dynamics and efficiency of DNA repair, in particular, after high LET radiation. When approaching this from another angle, heterochromatin has been shown to delay genome editing by clustered regularly interspaced short palindromic repeats-CRISPR associated protein 9 (CRISPR-Cas9) mutagenesis, which is highly dependent on DSB repair, but the outcome of mutagenic DNA repair is not influenced [48]. Future analysis using additional methods will further aid to clarify the roles of damage induction and repair after high and low LET damage, but based on the γH2AX-analysis, most of the gamma-induced damage was rapidly repaired within one hour. This limited need of the slow repair process may be why an improved DNA repair by open chromatin does not seem to be a major contributor in the response to gamma radiation.

In theory, our data would indicate a strong induction of DNA damage by high LET radiation even in heterochromatic cells. In support of this, there are reports of a better tumour control, apoptosis induction, and improved survival using carbon ions vs. X-ray therapy in a melanoma-bearing mouse [49]. The daughter cells of bronchial epithelial or non-small cell lung cancer cell populations irradiated with alpha or gamma radiation were recently investigated. Interestingly, the daughters of gamma-irradiated cells became radioresistant with an increase in H3K9me3 and the activity of several HDACs, while the radiosensitivity was unchanged in alpha-irradiated cells [50]. Treatment with vorinostat/SAHA eliminated the radioresistance phenotype in the daughter cells of gamma-irradiated cells [50].

Clinically, targeted alpha therapy is indeed regarded as promising due to its increased efficacy in tumour cells combined with reduced side effects in normal tissues [51]. In addition to the properties already mentioned, the helium nucleus structure of alpha particles gives them a superficial penetration and they may work for micrometastases and liquid cancers as well. Interestingly, alpha radiation impacts several of the 4 or 5 Rs important for radiotherapy as well as targets the hallmarks of cancer [52]. Those are exemplified here as more severe DNA damage which is difficult to repair (the first of the Rs), targeting the hallmarks “replicative immortality” and “resistance to cell death”, followed by the next R, reoxygenation, which concerns the advantageous properties of alpha particles towards hypoxic cells commonly present in solid cancers, where the hallmarks “inducing angiogenesis” and “activating invasion and metastasis” are relevant as targets [51].

The relatively small potentiation when using alpha radiation in gamma radiation-exposed cells is most likely due to two main factors: the cell cycle-specific effect of chromatin structure modifications and the mixed cell population when cells regrow after radiation exposure. The chromatin compactness normally varies throughout the cell cycle to allow for chromosome segregation and cell division. Generally, HP1γ siRNA slightly sensitises NSCLC cells to radiation exposure, and we reported that NSCLC TICs displayed increased levels of gamma radiation-induced micronuclei specifically at 50 h postexposure when using siRNA towards HP1γ, interpreted as during a key phase [25]. The second issue relates to the stem-like cell model systems which are challenging irrespective of enrichment method in the sense that they do not represent pure cell populations; stem cell marker-positive cells produce marker-negative progeny by asymmetric division, and effects get diluted.

With respect to the clinic, our results suggest that alpha emitters are potentially more efficient in cancers which recur with stem-like or heterochromatic properties, i.e., therapy-resistant cancers. Our results may indicate that HDACi should be avoided in connection with alpha emitters but suggest that HDACi pretreatment before photon radiotherapy may be beneficial for breast cancer patients with a basal subtype. The latter is supported by data on HDACi combined with other cytotoxic agents in breast cancer [53] and specifically using a novel HDACi YCW1 combined with IR in triple negative cells in culture and an orthotopic mouse model [54].

## 5. Conclusions

In conclusion, our results show that heterochromatin makes cells resistant to gamma radiation but less so to alpha radiation. We show that gamma radiation-exposed cells may be targeted using alpha radiation and provide a mechanistic basis for the involvement of chromatin in these effects.

## Figures and Tables

**Figure 1 cells-09-01165-f001:**
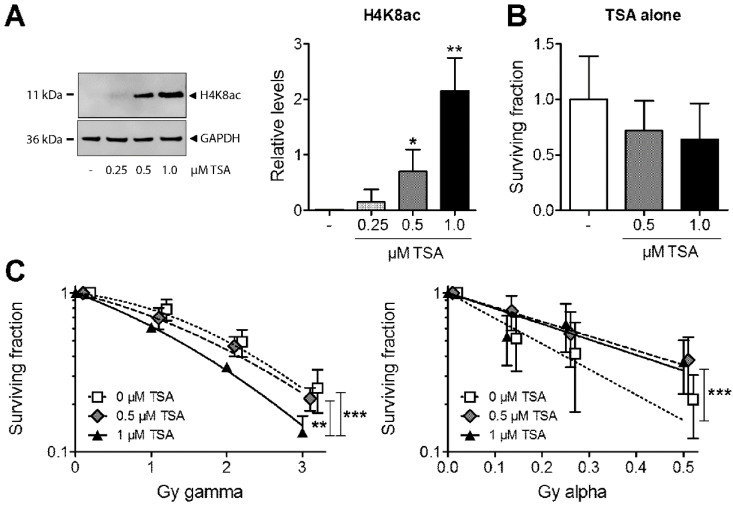
(**A**) Levels of acetylated lysine 8 of histone H4 (H4K8ac) were analysed in MDA-MB-231 cells using Western blot after treatment with 0.25–1 µM trichostatin A (TSA). Glyceraldehyde 3-phosphate dehydrogenase (GAPDH) was used as a loading control. (**B**) Effects of TSA alone on clonogenic survival using 0.5 and 1 µM TSA relative to untreated control cells. * *p* < 0.05 and ** *p* < 0.01, for 0.5 or 1 versus 0 µM TSA, respectively. (**C**) Clonogenic survival analysis of MDA-MB-231 cells pretreated with 0.5 or 1 µM TSA for 18 h before exposure to 1–3 Gy of gamma or 0.125–0.5 Gy of alpha radiation: All TSA groups were set to 1 for 0 Gy. ** *p* < 0.01 for 1 versus 0.5 µM TSA and *** *p* < 0.001 versus 0 µM TSA. *.

**Figure 2 cells-09-01165-f002:**
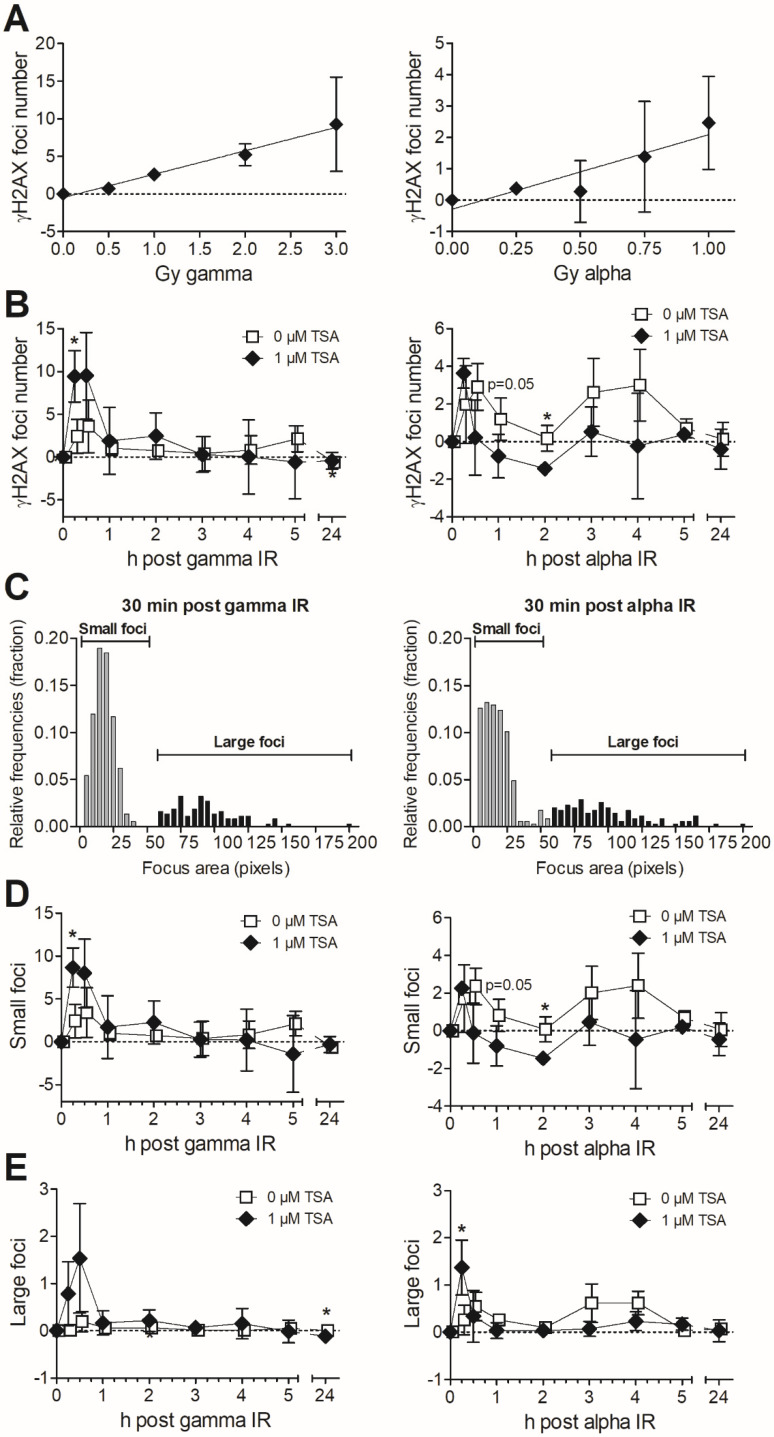
(**A**) The dose response of gamma and alpha radiation was evaluated at 30 min post-irradiation by analysis of the γH2AX foci number in MDA-MB-231 cells. (**B**) The repair kinetics of γH2AX foci are presented from 15 min up to 24 h postexposure to 2 Gy of gamma or 0.75 Gy of alpha radiation in MDA-MB-231 cells pretreated with 1 µM TSA for 18 h. (**C**) Focus areas per cell in pixels were plotted as a histogram, using the relative frequencies (where the sum is 1) on the Y-axis, to show the discrimination between small and large foci. Data was pooled from the 30 min time point of all experiments (0 and 1 µM TSA). The numbers of small (**D**) and large (**E**) foci are displayed, using the data from Figure 2C. The foci numbers for controls (0 Gy) were subtracted from sample foci numbers in all graphs to allow for comparisons between 0 and 1 µM TSA; therefore, negative values are also seen. * *p* < 0.05 versus 0 µM TSA.

**Figure 3 cells-09-01165-f003:**
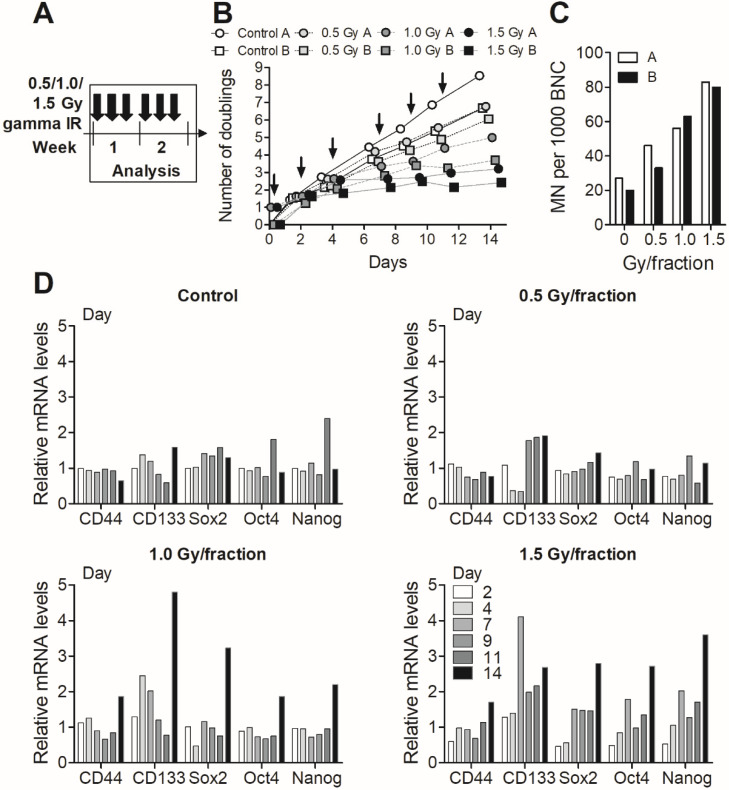
(**A**) Two parallel cell replicates of MDA-MB-231 (denoted A and B) were exposed to 6 × 0.5, 1.0, or 1.5 Gy fractions of gamma radiation, 3 days per week. (**B**) Cell growth was assayed and presented as number of doublings for 14 days, with the ionising radiation (IR) fractions indicated by arrows. (**C**) The frequency of micronuclei (MN) per 1000 binucleated cells (BNC) were scored at day 15 in replicate A and B. (**D**) The mRNA levels of cancer/normal stem cell markers CD44, CD133, Sox2, Oct4, and Nanog were analysed by real-time PCR at days 2, 4, 7, 9, 11, and 14 in replicate A.

**Figure 4 cells-09-01165-f004:**
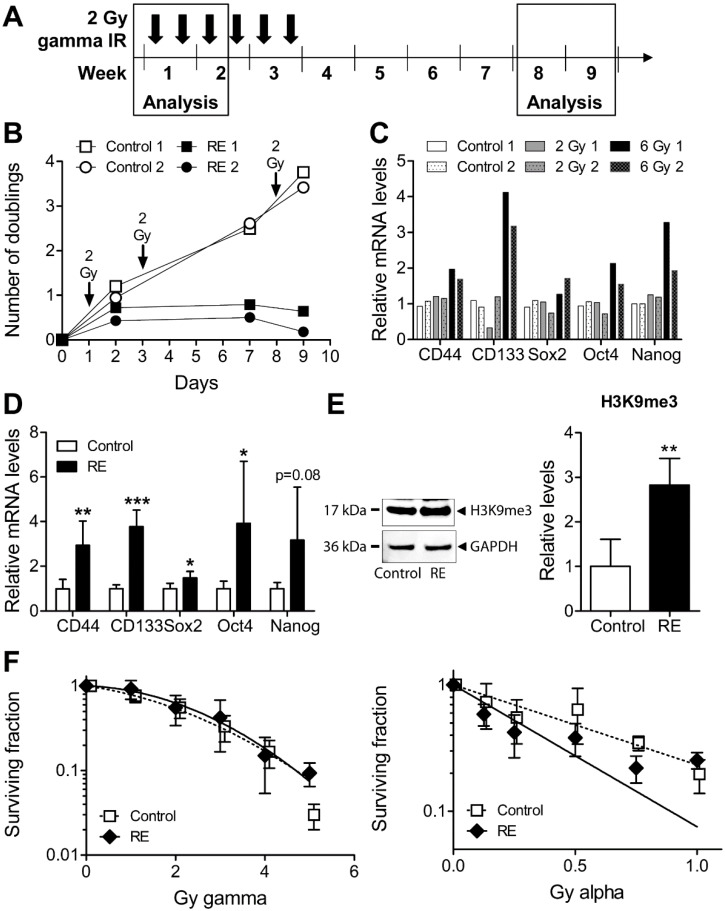
(**A**) Two parallel cell replicates (denoted 1 and 2) were exposed to 6 × 2 Gy fractions of gamma radiation, 2 days per week, aiming to establish gamma radiation-exposed (RE) MDA-MB-231 cells. Analysis was performed at early (2 replicates) and late time points (from replicate 1, three independent experiments). (**B**) Cell growth was assayed and presented as number of doublings during the first 9 days, with the 2 Gy fractions indicated as arrows. The mRNA levels of cancer/normal stem cell markers CD44, CD133, Sox2, Oct4, and Nanog were analysed by real-time PCR at 24 h after 2 Gy or in total 6 Gy, i.e., in samples collected on day 2 and 9 (**C**) or in samples 4–6 weeks after 12 Gy (**D**). Trimethylated lysine 9 of histone H3 (H3K9me3) levels, normalised to GAPDH, were assayed using Western blot (**E**), and clonogenic survival was analysed in RE cells versus control cells in response to gamma and alpha radiation in samples 4–6 weeks after 12 Gy (**F**). * *p* < 0.05, ** *p* < 0.01, and *** *p* < 0.001 for RE versus control cells.

**Figure 5 cells-09-01165-f005:**
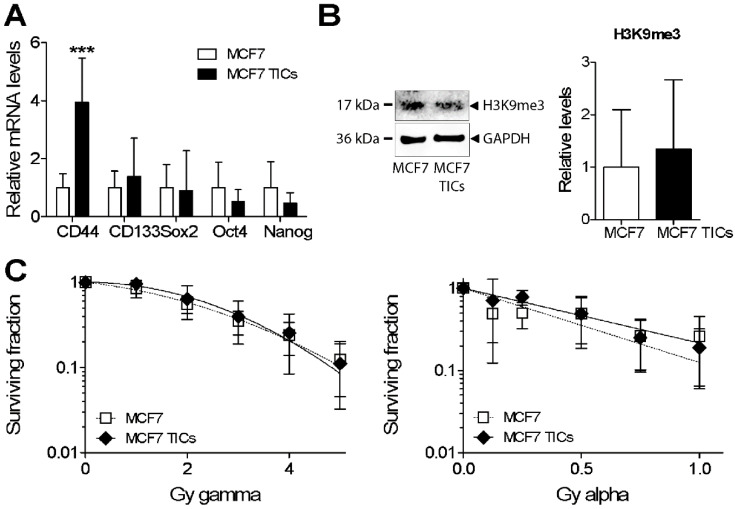
Sphere-forming MCF7 tumour-initiating cells (TICs) versus MCF7 were analysed at the level of CD44, CD133, Sox2, Oct4, and Nanog mRNA by (**A**) real-time PCR; (**B**) H3K9me3, normalised to GAPDH, was assayed using Western blot; and (**C**) clonogenic survival was analysed in MCF7 TICs versus MCF7 cells in response to gamma and alpha radiation. *** *p* < 0.001 for MCF7 TICs versus MCF7 cells.

**Figure 6 cells-09-01165-f006:**
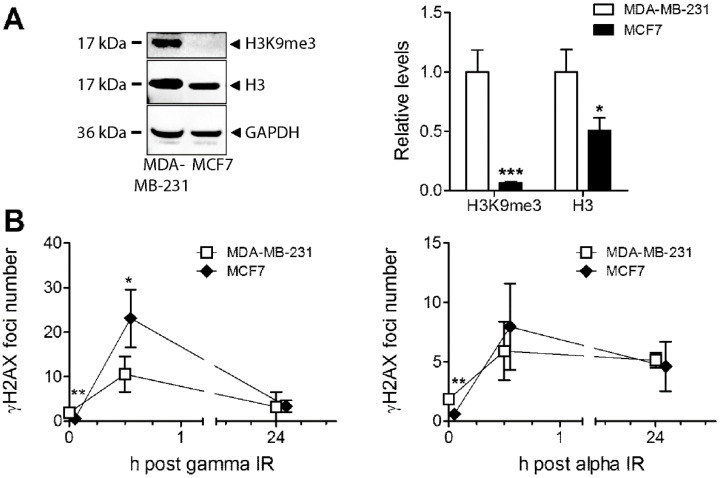
(**A**) Basal levels of H3K9me3 and total protein levels of histone H3 were analysed in MDA-MB-231 and MCF7 cells using Western blot. GAPDH was used as a loading control. (**B**) γH2AX foci numbers are presented at 30 min and 24 h postexposure to 6 Gy of gamma or 2 Gy of alpha radiation in MDA-MB-231 and MCF7 cells. * *p* < 0.05, ** *p* < 0.01, and *** *p* < 0.001 versus MDA-MB-231.

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
