# Peer review of "Alpha Radiation as a Way to Target Heterochromatic and Gamma Radiation-Exposed Breast Cancer Cells"

_cells, 2020, doi:10.3390/cells9051165_

Round 1

Reviewer 1 Report

In this study, the authors have analysed cell survival and DNA damage induction (via γH2AX foci) in breast cancer cells following α-particle and γ-irradiation (high and low linear energy transfer (LET), respectively) in the absence and presence of the histone deacetylate inhibitor, trichostatin A (TSA). This was aimed at investigating whether chromatin compaction has an impact on cell radiosensitivity through increases in DNA damage formation or repair. It was found that TSA caused increased sensitivity of MDA-MB-231 cells to low-LET γ-rays with an associated early induction in γH2AX foci, but that there was increased resistance to high-LET α-particles, with relatively little changes in DNA damage induction and repair. Furthermore using fractionated γ-radiation, the authors suggest that this induces a more tightly packed chromatin with an increase in stem cell markers, which caused an elevated sensitivity to α-particle radiation only.

In my opinion, there are some major limitations in this study with the inclusion of some extremely variable data which make this too preliminary at this stage for publication. There are also a number of supporting experiments which need to be performed to strengthen the data, of which I’ve detailed in the comments below.

Specific comments:

  1. The authors exclusively use analysis of histone markers, particularly H4K8 acetylation and H3K9 trimethylation as markers of euchromatin and heterochromatin. There needs to be additional supporting evidence (e.g. MNase digestion) that chromatin compaction is altering following both TSA treatment and in response to fractionated radiation, which would support that this is a major factor controlling cell survival in response to low/high-LET radiation.
  2. There is a concern that the results from clonogenic assay experiments are extremely variable (particularly Figures 1C, 4F and 5C), making the data very difficult to interpret. The Methods section suggests that manual counting was performed on the assays, but I would recommend that the authors use an automated system which may provide more consistency and accuracy. It is also noticeable that the majority of these experiments have been performed over a limited one log fold change in survival, when in fact significantly increased doses of radiation could have been used.
  3. Appearance of γH2AX foci is used as a marker of DNA double strand breaks (Figure 2) in an attempt to correlate changes in survival with DNA damage repair. As I’m sure the authors are aware, γH2AX is used as a surrogate marker of the initial signalling events leading to the recognition and repair of double strand breaks. Therefore, it is recommended that neutral comet assays be performed which are a direct measurement of the DNA damage itself to provide further evidence that DNA repair kinetics are altered in the presence of TSA. Nevertheless, whilst there is a strong induction in γH2AX foci following g-radiation in the presence of TSA which would correlate with decreased survival, the changes in foci formation following α-particle irradiation are fairly minimal. There is no convincing or compelling evidence therefore that the cells are able to repair the more complex DNA damage induced under these conditions that promotes cellular resistance. Potentially the authors could examine complex/clustered DNA damage directly (e.g. enzyme modified comet assays or pulsed-field gel electrophoresis) which would add support to this hypothesis.
  4. It would also be interesting to supplement the γH2AX foci data with analysis of 53BP1 and RAD51 foci as the downstream markers of non-homologous end-joining (NHEJ) and homologous recombination (HR), respectively. This would be of value given that there exists evidence that the repair of complex/clustered DNA damage induced by high-LET radiation may be preferentially repaired by HR.
  5. There is no statistical analysis of the expression of the stem cell marker data (Figure 3C, 4C), so it is difficult to interpret the significance of these. The Methods would suggest that most likely these data were only from 1-2 experiments, and therefore these have to be taken at face value. Additionally, it is suggested that the protein levels of these stem cell markers are also analysed to support the mRNA data, at least from the key experiments (Figure 4D, 5A).
  6. The increase in H3K9 trimethylation in radiation-exposed cells (Figure 4E) visually looks almost exactly the same as the control cells, and doesn’t’ really reflect the 3-fold changes shown in the graph.

Author Response

We would like to express our sincere gratitude to the reviewer for the helpful comments and suggestions of complementing experiments. We agree that it would be very interesting to go further into details on several of the points, and hope to be able to do so in future projects. But to include such a number of complementary experiments (MNase digestion, higher doses for clonogenic survival, neutral comet assay, enzyme modified comet assays or pulsed-field gel electrophoresis, 53BP1 and RAD51 foci, 1-2 more experiments in Fig 3C and 4C, protein levels of stem cell markers) in the current manuscript would be very difficult, especially since, the time frame given for this revision is 10 days. Therefore, we will restrict the revision to changes in the manuscript plus some additional analyses and hope for the understanding of the reviewer.

Below we address the specific comments, in particular the concerns on variability and supporting experiments:

  1. The H4K8 modification has previously been shown to be induced by TSA and was used as a confirmatory measure to validate that chromatin was open at the time of radiation exposure. Both H4K8 acetylation and H3K9 methylation have previously been reported to correlate positively or negatively, respectively, with MNase digestion (Yang et al., PMID: 20718950). It would be interesting to add MNase digestion in a future experiment for the analysis of gamma-radiation exposed cells, we have a sentence on this at the end of the first paragraph of the discussion: In future studies, it would also be of value to assess changes in chromatin structure using methods such as micrococcal nuclease assay or DNA staining.
  2. We are aware of the variability within certain experiments, in particular for the clonogenic survival after exposure to alpha radiation. One general feature of alpha exposure is that not all cells are hit at lower doses (on average 2-4 tracks/cell at 1 Gy) and we used doses from 0.125 up to 0.5/1 Gy here, which per se creates a larger variability than for gamma radiation where ionisation events are more evenly distributed. The exposure procedure is also more complex for alpha than for gamma exposure, for the latter whole plates are placed inside the machine for exposure. To get cells close enough to the alpha source, cells are grown on coverslips that are transferred to special discs which are covered with a thin plastic film (Mylar foil) before exposure. After exposure, the foil is removed, and the coverslips are transferred back to the 6-well plate. This removal of Mylar foil causes cell loss, in particular when cells are anyway stressed by being plated sparsely as for the clonogenic survival.

We choose to plate the same number of cells for all doses, with higher numbers for alpha exposure than gamma exposure. This is described as follows in M&M: Cells were plated at a density of 200 cells/well for gamma radiation and 400 cells/well for alpha radiation (due to cell loss when removing the Mylar foil). MCF7 TICs were seeded at 400 or 800 cells/well, respectively. We chose this type of approach since by having the same initial cell number, the removal of Mylar would be expected to cause a similar stress for cells on all coverslips. While having the advantage of “equal stress” by the Mylar, the disadvantage with this approach is that when few colonies are formed (at higher doses), the standard deviation between experiments becomes larger.

The alternative approach is to plate a larger number of cells for the higher doses (for example 10 times more cells if a surviving fraction of 0.1 is expected at a certain dose), to get higher numbers of colonies there in the end and improved statistics. The disadvantage is that the cell loss by the Mylar may be reduced due to the much larger cell number at start, making the colony numbers not easily comparable. MDA-MB-231 cells are perhaps more sensitive to the Mylar removal than other cell lines where alpha exposure was performed previously for clonogenic survival. Some labs also have the opposite direction of the source, and instead grow cells on Mylar, which is placed on the source.

In retrospect, it is difficult to say which approach would have caused the least variability. To compensate partly, we made a larger number of experiments and used the average of all experiments, and (except for most of the very early experiments for Fig 1C), made duplicate slides for each experiment to average out differences within an experiment.

Our selected approach was also a reason for not going too high in dose, since this inherently means a very low number of colonies on the plates and an even larger experimental variation. For Figure 4F and 5C we still went up to 1 Gy, but to go higher (over one log fold change in survival) would have meant to change the chosen approach and repeat everything.

In response to the suggestion on automated scoring, we have now scanned all plates for Figure 1C and analysed them using semi-automated scoring (with the software already described in the manuscript). We have attached the comparison of manual and automated graphs to this revision for review purpose. The patterns are relatively similar using both approaches, but the variability is not reduced, rather the contrary. Automatic scoring is advantageous in many cases and works well for good quality plates with clear and sharp colonies. Among this set of plates we had some with colonies where the central part fell off during staining, some weak colonies that were not detected well, and in some cases aggregates of colonies. Those are usually better detected by manual scoring. This problem with is further described in (Brzozowska et al. PMID: 30673853).

These sentences were added to the discussion in response to the comments in 2-4:

The results using alpha radiation are more variable, due to reasons such as a more challenging exposure procedure and the inherent properties of alpha radiation where not all cells are hit at lower doses. Future studies using other methods are needed, still the pattern after alpha radiation is clearly different from that after gamma radiation. It would also be interesting to analyse downstream markers of NHEJ, such as 53BP1 foci, and homologous recombination (HR, RAD51 foci), to further pinpoint a possible preferential repair pathway for high LET-induced complex/clustered DNA damage.

  1. Yes, the neutral comet assay could have been a helpful assay for direct measurement of DNA damage. We tried the neutral comet assay in previous experiments with another cell line of similar radiation sensitivity, but were only able to detect clear differences to background signal after even higher radiation doses (5-10 Gy) than those used here (up to 3 Gy). So, in our hands, it was technically not possible to use it in the dose ranges that were realistic in terms of survival. We have successfully used the alkaline comet assay in previous publications, but it is not specific for double strand breaks. Previous publications also report on a higher sensitivity of the alkaline versus the neutral comet assay (Lu et al. PMID: 29053680). We did try Fpg- and Endo III-modified comet assays as well previously, without success unfortunately, and do not have access to pulsed-field gel electrophoresis equipment.

We agree that the data for clonogenic survival and gammaH2AX are more obvious after gamma radiation exposure, but the response to alpha radiation is clearly different than that for gamma radiation and shows the opposite trend, based on the averages of our experiments. We changed to “tended to” improve in the abstract and added “may” indicate in the last paragraph of the discussion, and added a sentence in the discussion commenting this (see above).

  1. This would also be an interesting future study, comments are added as well to the discussion (see above).
  2. As described in the Materials and methods in section 2.8 and the figure legends, these data are from 1-2 experiments (replicate A and B in Fig 3A-B = 2 experiments, replicate A in Fig 3C = 1 experiment, and replicate 1 and 2 in Figure 4B-C = 2 experiments): “Results are displayed as mean +/- standard deviation from at least 3 experiments, except for Figure 3 and 4B-C where 1-2 replicates are shown.”

The reason for this was that these were planned as preliminary experiments to set an appropriate dose level and number of fractions, where the continued or follow up-experiments were performed using 3 separate experiments (Fig 4D-F). Later on, we realised that it would be of value to also show these preliminary/preparatory data despite the lack of statistical significance, because it gives an indication of the progression of the phenotype at an early time point. This is the reason for this perhaps unusual presentation of data, but the setup for testing fractionation schemes is clearly described in 3.3.

When it comes to the analysis of stem cell markers at the protein level, this is also rather demanding due to few positive cells and very low levels of stem cell markers in normal (control) cell lines (except perhaps CD44). We chose to stay with this method since the detection level (sensitivity) and accuracy for real time PCR analysis is generally much higher than what can be achieved with most antibodies for Western blot, when it comes to these stem cell markers. The increase in stem cell marker expression correlates with a poor prognosis in the clinic, but the protein expression is still relatively low and usually displays a patched pattern which is less easy to quantify, as displayed by immunofluorescence in Yang et al. (PMID: 30464534).

6. We improved the exposure generally in this western image, the difference may now appear better visually. The other blots had a more unequal loading and were therefore less visual to display, in our opinion.

Reviewer 2 Report

The manuscript aims at investigating DNA damage induction and repair in the context of chromatin structure for alpha and gamma radiation, also investigating potential therapeutic applications.

The authors wrote the manuscript in a very clear and concise manner. The experimental design was well established, and the results support the conclusions. The only suggestion I would give is to discuss in more details the results comparing to available data in the literature, even with Monte-Carlo simulation studies (i.e. https://doi.org/10.1002/mp.13405).

Therefore, I strongly recommend this manuscript for publication after appreciation by the authors regarding the suggestion given here.

Author Response

Thank you very much for the positive feedback on our manuscript. We have now included some more general discussion on this topic, as well as discussion on the specific reference.

Round 2

Reviewer 1 Report

I very much appreciate the authors very detailed response to my comments. Despite this, the additional experiments which I feel are necessary for improving the manuscript have not been completed and included in the revision. Therefore, the authors should take the amount of time necessary to complete these.

I accept the variability in the clonogenic assay data with alpha radiation, but the restrictive doses used over a just one log fold change in survival is a major limitation that needs to be addressed. I also feel strongly that the manuscript would benefit greatly from key experiments, particularly MNase digestion to examine chromatin compaction (which is a relatively simple experiment) and comet assay analysis for double strand breaks and complex damage (if the authors have been unsuccessful in this approach, then seeking further support from other experts would be advised). Supplementation of the data with 53BP1 and RAD51 foci analysis would be interesting, although are not essential. I accept that the authors have adequately responded to my other major comments.

Reviewer 3 Report

The authors addressed most of my comments.